# Learning to Clean: Reinforcement Learning for Noisy Label Correction

**Marzi Heidari[1], Hanping Zhang[1], Yuhong Guo[1,2]**
[1]School of Computer Science, Carleton University, Ottawa, Canada
[2]CIFAR AI Chair, Amii, Canada
{marziheidari@cmail, jagzhang@cmail, yuhong.guo@}.carleton.ca

## Abstract

The challenge of learning with noisy labels is significant in machine learning, as it can severely degrade the performance of prediction models if not addressed properly. This paper introduces a novel framework that conceptualizes noisy label correction as a reinforcement learning (RL) problem. The proposed approach, Reinforcement Learning for Noisy Label Correction (RLNLC), defines a comprehensive state space representing data and their associated labels, an action space that indicates possible label corrections, and a reward mechanism that evaluates the efficacy of label corrections. RLNLC learns a deep feature representation based policy network to perform label correction through reinforcement learning, utilizing an actor-critic method. The learned policy is subsequently deployed to iteratively correct noisy training labels and facilitate the training of the prediction model. The effectiveness of RLNLC is demonstrated through extensive experiments on multiple benchmark datasets, where it consistently outperforms existing state-of-the-art techniques for learning with noisy labels.

## 1 Introduction

Deep neural networks have made significant strides in various domains of machine learning [1–4]. These models predominantly utilize supervised learning, which depends extensively on the availability of large datasets with high-quality labeled data. Unfortunately, in real-world applications, the quality of labels is frequently compromised by various issues, including subjective errors from indistinguishable samples and objective mistakes in label recording, making noisy labels a common challenge. The presence of such label noise has been shown to significantly undermine the generalization ability of deep neural networks. Therefore, the development of robust methods for learning from noisy labels is of paramount importance [5].

Label noise has been categorized into two primary types: class-conditional noise (CCN) and instance-dependent noise (IDN) [6], while many recent works have focused on the more common IDN, where noise distribution correlates with feature similarities among samples [7–9]. Deep neural networks tend to learn from clean data initially and subsequently adapt to noisy data, which underpins several sample selection methods that utilize training dynamics, such as prediction loss and confidence levels, to isolate and manage noisy labels [10, 11]. To enhance the resilience of models trained on noisy datasets, other methods have evolved to include sophisticated training regimens that incorporate label filtering [12], label correction [13–15], robust loss formulations [16], and semi-supervised learning frameworks [17, 18]. However, these methods typically lack the ability to actively explore different possibilities or learn from long-term consequences, which can limit their effectiveness in more complex or highly noisy scenarios.

To address this drawback, we propose to formulate the task of noisy label rectification as a reinforcement learning problem. Reinforcement learning can dynamically adapt its noisy label correction

39th Conference on Neural Information Processing Systems (NeurIPS 2025).

strategy based on feedback from its actions, optimizing performance through a sequence of decisions that maximize a cumulative, long-term reward. This ability to make sequential, non-myopic decisions is particularly well-suited for addressing label noise in complex and instance-dependent environments. Specifically, we devise a novel policy gradient method, named as Reinforcement Learning for Noisy Label Correction (RLNLC), to learn a stochastic policy function, which determines the label correction actions in a given state and consequently separates the training data into a clean subset without label corrections and a noisy subset with corrections, by maximizing the expected cumulative reward. A key component of this RLNLC method is the reward function, which evaluates the impact of actions and the resulting states. We design the reward function to enhance both label consistency across the entire dataset and the inter-subset alignment of corrected labels with clean labels. This is achieved through k-nearest neighbor prediction mechanisms, aiming to enhance the robustness of label correction, maintain instance-dependent and prediction-aware label smoothness, and support the performance of downstream label prediction tasks. This RL formulation allows for dynamic adaptation to the evolving data characteristics, offering a robust method for noise correction. We evaluate RLNLC through rigorous experiments on benchmark datasets, demonstrating its effectiveness in various noisy settings. The contributions of this work are summarized as follows:

- We innovatively formulate noisy label correction as an RL problem and propose RLNLC, a method that leverages the adaptive capabilities of RL to effectively address label noise.

- We design a tailored policy function based on a deep representation network, which adaptively determines label correction actions by considering the complete state of the data.

- We develop an informative reward function that encourages effective label correction by aligning with both local data structures and confident clean labels through k-nearest neighbor prediction mechanisms.

- We devise an efficient encoding scheme for the critic network to enable effective deployment of the actor-critic framework for optimizing the policy function.

- Extensive experiments demonstrate that RLNLC outperforms existing state-of-the-art methods, highlighting the effectiveness of this innovative RL approach.

## 2 Related Work

### 2.1 Learning with Noisy Labels

Deep neural networks are highly susceptible to overfitting when trained on datasets with noisy labels and thus perform poorly on clean test datasets [5]. Previous studies have primarily focused on two types of label noise, class-conditional label noise (CCN) and instance-dependent label noise (IDN).

**Class-Conditional Label Noise** For the CCN label noise, the probability of a label being noisy is conditional on the class but not on individual instances. To address CCN, MentorNet [10] and Co-teaching [11] use networks to select low-loss, reliable samples through guidance. Reweighting techniques adjust the training influence of each sample based on estimated noise levels, thus enhancing robustness [19]. Decoupling [20] updates model parameters only on instances with classifier disagreement, thus mitigating the reinforcement of noisy labels. Nested Dropout [21] integrates dropout with curriculum learning, increasing sample difficulty progressively to enhance model resilience to noise. Semi-supervised learning strategies, e.g., DivideMix [17], use loss distribution modeling to differentiate between clean and noisy data, treating noisy labels as unlabeled to leverage semi-supervised techniques.

**Instance-Dependent Label Noise** In the IDN scenario, the label noise probability varies with instance features, presenting unique challenges. To address this problem, CleanNet [22] leverages a clean validation set to assess and correct noisy labels accurately. Advanced strategies like LongReMix [18] combine consistency regularization with data augmentation to effectively handle the variability in noise. Pseudo-Label Correction (PLC) [15] refines labels during training using the model's predictions, enhancing label accuracy. Adaptive reweighting methods such as BARE [23] dynamically adjust the weights of samples based on their likelihood of being clean, addressing the complexities of IDN. SSR [24] employs statistical methods to estimate and correct noise rates, facilitating more effective learning from noisy datasets by adapting training strategies to the specific

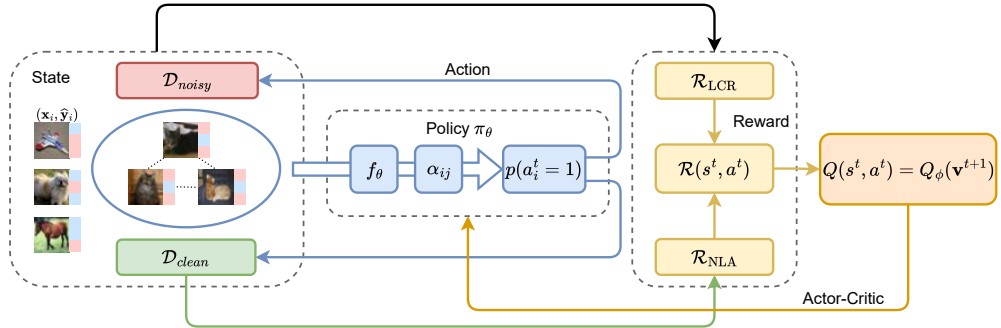

Figure 1: Overview of the proposed RLNLC. Each data point $\mathbf{x}_i$ is associated with an initial label $\widehat{\mathbf{y}}_i$ that is potentially noisy. The policy network $\pi_\theta$ is constructed over a deep feature extraction network $f_\theta$, and it determines actions based on the current state of the data $\boldsymbol{s}^t = \{(\mathbf{x}_i, \widehat{\mathbf{y}}_i^t)\}_{i=1}^N$, resulting in label corrections. The updated labels subsequently lead to the next state. The reward function is designed to evaluate the labels in an instance-dependent manner, capturing dataset-wide label consistency and the inter-subset alignment of the noisy labels with clean labels. The policy function is learned using an actor-critic method.

characteristics of the noise. SURE [25] enhances uncertainty estimation by integrating techniques from model regularization like RegMixup, cosine similarity classifier, and sharpness aware minimization to tackle IDN.

## 2.2 Reinforcement Learning

Reinforcement learning (RL) is a machine learning framework in which an intelligent agent learns to make optimal decisions by interacting with its environment and achieving goals through sequential decision-making [26]. RL is known for its ability to explore effectively [27, 28] and to learn optimized decisions for long-term outcomes [29, 30]. Leveraging these strengths, RL has seen extensive application in decision-making domains over the past few years [31, 32]. In particular, the remarkable success of ChatGPT has provoked significant interest in Reinforcement Learning from Human Feedback (RLHF) [33–35], where RLHF fine-tunes complex LLMs by leveraging a reward model trained on human-provided feedback. Recent work has applied RLHF to fine-tune text-to-image diffusion models, achieving superior alignment with user preferences [36]. Another study reformulated semi-supervised learning (SSL) as a bandit problem to guide classifier training using weighted rewards [37]. More recently, photo-finishing tuning has been framed as a goal-conditioned RL problem, allowing iterative optimization of pipeline parameters guided by a goal image [38]. These advancements highlight RL's adaptability in fine-tuning and optimizing diverse machine learning systems.

**Policy Gradient Methods**   Policy gradient is a class of RL methods designed to directly learn an optimal policy that maximizes cumulative rewards. REINFORCE [39], a foundational policy gradient method, uses Monte Carlo simulations to estimate the value function and calculate the policy gradient. Actor-Critic methods [40] extend this by incorporating a value function (the Critic) to evaluate the policy (the Actor), providing feedback to assist in updating the policy. One of the key advantages of policy gradient methods is their ability to learn stochastic policies, making them effective in complex environments requiring exploration. Additionally, they are well-suited for high-dimensional or complex action spaces, driving their adoption across diverse applications [41, 36, 37].

## 3   Method

### 3.1   Problem Setup

We assume a noisy classification training dataset $\mathcal{D} = (X, \widehat{Y}) = \{(\mathbf{x}_i, \widehat{\mathbf{y}}_i)\}_{i=1}^N$, where each input instance $\mathbf{x}_i \in \mathcal{X}$ is paired with an observed label vector $\widehat{\mathbf{y}}_i \in \mathcal{Y}$ that is potentially a corrupted noisy version of the true label vector $\mathbf{y}_i$. The objective is to learn an effective prediction model, defined as

the composition $h_\psi \circ f_\theta : \mathcal{X} \to \mathcal{Y}$, where $f_\theta$ denotes the feature extractor with parameters $\theta$, and $h_\psi$ represents the classifier with parameters $\psi$. We aim to develop an effective label cleaning technique to correct noisy labels and enable efficient training of prediction models without being hindered by the challenges posed by label noise.

## 3.2 RL for Noisy Label Correction

Reinforcement Learning (RL) problem is often modeled as a Markov Decision Process (MDP) [26], characterized by the tuple $\mathcal{M} = (\mathcal{S}, \mathcal{A}, P, \mathcal{R}, \gamma)$, where $\mathcal{S}$ denotes the state space, $\mathcal{A}$ represents the action space, $P(s'|s, a)$ defines the transition dynamics for $s, s' \in \mathcal{S}$ and $a \in \mathcal{A}$, $\mathcal{R} : \mathcal{S} \times \mathcal{A} \to \mathbb{R}$ is a reward function, and $\gamma \in (0, 1)$ is the discount factor. The primary objective is to determine an optimal policy $\pi^\star : \mathcal{S} \to \mathcal{A}$ that maximizes the expected discounted cumulative reward $J_r(\pi) = \mathbb{E}_\pi[\sum_{t=0}^\infty \gamma^t \mathcal{R}(s^t, a^t)]$.

In this work, we formulate the challenge of noisy label cleaning as an RL problem by properly designing and defining the key components of the MDP. We then deploy an actor-critic method with state encoding to learn an optimal policy function for noisy label correction. The overall framework of the proposed RLNLC is illustrated in Figure 1, with the approach detailed in the remaining section.

### 3.2.1 State

We define the state as the observed data and corresponding (corrected) labels, expressed as $s^t = \mathcal{D}^t = (X, \widehat{Y}^t) = \{(\mathbf{x}_i, \widehat{\mathbf{y}}_i^t)\}_{i=1}^N$ at any time-step $t$. The provided training dataset $\mathcal{D} = (X, \widehat{Y})$ can be treated as a static initial state $s_0^0$. To enhance the exploratory aspect of our RL methodology, we initiate each RL process by randomly altering a small subset of labels in $s_0^0$ to establish an initial state $s^0$. The state encapsulates the current knowledge about the environment and serves as the input for the policy function $\pi$. As the model operates, the policy $\pi$ selects actions based on the current state $s^t$. These actions primarily involve modifying the labels, which leads to a transition to a new state, $s^{t+1} = (X, \widehat{Y}^{t+1})$. Ideally, we expect to reach an optimal goal state where the labels are "clean".

### 3.2.2 Action and Policy Function

We define the action space $\mathcal{A}$ as the possible binary valued vectors of label correction decisions on all the data instances in any state, such that for each action vector $a = [a_1, \cdots, a_i, \cdots, a_N] \in \mathcal{A}$, $a_i \in \{0, 1\}$ indicates whether the current label for instance $\mathbf{x}_i$ needs to be corrected ($a_i = 1$).

At the core of RLNLC is a probabilistic policy function, $\pi_\theta$, which maps a given state, $s^t$, to actions in a stochastic manner. Given the state and action definitions above, the policy function is designed to make probabilistic decisions regarding whether to apply label corrections to each instance $\mathbf{x}_i$ in the dataset (i.e., setting $a_i = 1$) given the current state $s^t = \{(\mathbf{x}_i, \widehat{\mathbf{y}}_i^t)\}_{i=1}^N$. To effectively identify and rectify noisy labels, we define a parametric policy function $\pi_\theta$ based on the label consistency of k-nearest neighbors, calculated within the embedding space generated by a deep feature extraction network $f_\theta$, which is initially pre-trained on the given training dataset $\mathcal{D}$ using standard cross-entropy loss. The underlying assumption is that a label inconsistent with those of its k-nearest neighbors is more likely to be noisy and in need of correction.

Let $\mathcal{N}(\mathbf{x}_i)$ denotes the indices of the k-nearest neighbors of instance $\mathbf{x}_i$ within the dataset $\mathcal{D}$ based on the Euclidean distances calculated in the embedding space, $f_\theta(\mathcal{X})$, extracted by the current $f_\theta$. For each instance $\mathbf{x}_i$, we then aggregate the labels of its k-nearest neighbors in the current state $s^t$ using an attention mechanism to generate a new label prediction, $\bar{\mathbf{y}}_i$, that aligns with the local data structures encoded by $f_\theta$. Specifically, $\bar{\mathbf{y}}_i$ is computed as follows:

$$\bar{\mathbf{y}}_i = \sum\nolimits_{j \in \mathcal{N}(\mathbf{x}_i)} \alpha_{ij} \widehat{\mathbf{y}}_j^t. \tag{1}$$

The attention weights $\{\alpha_{ij}\}$ are computed using similarities of instance pairs in the embedding space, such that:

$$\alpha_{ij} = \frac{\exp\left(\mathrm{sim}\left(f_\theta(\mathbf{x}_i), f_\theta(\mathbf{x}_j)\right)/\tau\right)}{\sum_{j' \in \mathcal{N}(\mathbf{x}_i)} \exp\left(\mathrm{sim}\left(f_\theta(\mathbf{x}_i), f_\theta(\mathbf{x}_{j'})\right)/\tau\right)}, \tag{2}$$

where $\tau$ is a temperature hyperparameter, and $\mathrm{sim}(.,.)$ denotes the cosine similarity. Finally the probability of taking each element action $a_i^t = 1$ for instance $\mathbf{x}_i$ in state $s^t$—and thus the policy

function—is computed by comparing the k-nearest neighbor predicted label vector $\bar{\mathbf{y}}_i$ with the current label vector $\widehat{\mathbf{y}}_i^t$, such that:

$$\pi_\theta(\boldsymbol{s}^t)_i = p(a_i^t = 1) = \frac{\sum_{j=1}^{C} \mathbb{1}(\bar{\mathbf{y}}_{ij} > \bar{\mathbf{y}}_{i\widehat{y}_i}) \cdot \bar{\mathbf{y}}_{ij}}{\sum_{j=1}^{C} \mathbb{1}(\bar{\mathbf{y}}_{ij} \geq \bar{\mathbf{y}}_{i\widehat{y}_i}) \cdot \bar{\mathbf{y}}_{ij}}, \tag{3}$$

where $C$ denotes the length of the label vector—the number of classes for classification, $\widehat{y}_i = \arg\max_j \widehat{\mathbf{y}}_{ij}$ denotes the original predicted class index for instance $\mathbf{x}_i$ in state $\boldsymbol{s}^t$.

The probability $p(a_i^t = 1)$ defined above quantifies the level of label inconsistency—specifically, the extent to which the class prediction from the k-nearest neighbors disagrees with the original class prediction $\widehat{y}_i$, thereby indicating the likelihood of noise and the probability of applying label correction to $\mathbf{x}_i$. To interpret, the probability $p(a_i^t = 1)$ is proportional to the sum of probabilities of classes that are more likely than the original predicted class $\widehat{y}_i$ in the k-nearest neighbor prediction $\bar{\mathbf{y}}_i$. This sum is then normalized by the sum of probabilities of classes whose probabilities are no less than that of the original label $\widehat{y}_i$ in $\bar{\mathbf{y}}_i$. This normalization ensures that $p(a_i^t = 1)$ scales appropriately relative to the level of disagreement in class predictions, without being affected by disagreement-irrelevant classes whose probabilities in $\bar{\mathbf{y}}_i$ are lower than that of $\widehat{y}_i$. Note, the probability $p(a_i^t = 1)$ becomes zero when all the other classes except $\widehat{y}_i$ become disagreement-irrelevant—i.e., class $\widehat{y}_i$ has the largest probability in $\bar{\mathbf{y}}_i$ and there is no prediction disagreement.

**Deterministic Transition with Stochastic Policy**  Given state $\boldsymbol{s}^t$, we determine the action vector $\boldsymbol{a}^t$ using the stochastic policy function $\pi_\theta$ introduced above, enhancing the exploratory behavior of the learning process. Specifically, we randomly sample the value for each binary-valued action element $a_i^t$ from a Bernoulli distribution, Bernoulli$(p_i)$, with $p_i = p(a_i^t = 1)$. Given $(\boldsymbol{s}^t, \boldsymbol{a}^t)$, we then deploy a deterministic transition model to induce the next state $\boldsymbol{s}^{t+1} = \{(\mathbf{x}_i, \widehat{\mathbf{y}}_i^{t+1})\}_{i=1}^N$, such that:

$$\widehat{\mathbf{y}}_i^{t+1} = \begin{cases} \widehat{\mathbf{y}}_i^t & \text{if } a_i^t = 0, \\ \bar{\mathbf{y}}_i & \text{if } a_i^t = 1. \end{cases} \tag{4}$$

This mechanism maintains a soft label distribution in vector space, ensuring that label corrections occur probabilistically based on the likelihood of a label being noisy. By introducing randomness, the learning process explores a broader range of state-action pairs, increasing the chances of discovering better strategies while avoiding suboptimal solutions. Additionally, it mitigates disruptions from abrupt label changes, allowing the model to adaptively learn from the evolving environment characterized by different states, ultimately enhancing the overall robustness of the learning system.

### 3.2.3  Reward Function

In RL, the reward function $\mathcal{R}(\boldsymbol{s}^t, \boldsymbol{a}^t)$ provides feedback on the quality of an action $\boldsymbol{a}^t$ taken in a given state $\boldsymbol{s}^t$, guiding the learning process toward its objective. To achieve the goal of cleaning label noise for the proposed RLNLC, we design the reward function to evaluate the quality of an action through the labels produced from taking the action. Our reward function $\mathcal{R}$ consists of two sub-reward evaluation functions: a label consistency reward function, $\mathcal{R}_{\text{LCR}}$, and a noisy label alignment reward function, $\mathcal{R}_{\text{NLA}}$.

**Label Consistency Reward**  With the deterministic transition mechanism, the next state $\boldsymbol{s}^{t+1}$ is obtained in a fixed manner by taking action $\boldsymbol{a}^t$ in the given state $\boldsymbol{s}^t$. Consequently, the quality of the labels $\{\widehat{\mathbf{y}}_i^{t+1}\}_{i=1}^N$ in $\boldsymbol{s}^{t+1}$ directly reflects the quality of the state-action pair $(\boldsymbol{s}^t, \boldsymbol{a}^t)$. We design the label consistency reward (LCR) function, $\mathcal{R}_{\text{LCR}}$, to assess the label quality in $\boldsymbol{s}^{t+1}$—i.e., how well each label fits the dataset—based on the k-nearest neighbor label prediction mechanism. To separate the impact of the policy function from label quality evaluation, we employ a fixed backbone model, $f_\omega$, pre-trained on the original dataset to extract embedding features for the data instances in $\boldsymbol{s}^{t+1}$, supporting k-nearest neighbor label prediction. Specifically, the label consistency reward function quantifies the negative Kullback-Leibler (KL) divergences between the given labels in $\boldsymbol{s}^{t+1}$ and the k-nearest neighbor predicted labels across all the $N$ instances, such that:

$$\mathcal{R}_{\text{LCR}}(\boldsymbol{s}^t, \boldsymbol{a}^t) = -\mathbb{E}_{i \in [1:N]} \left[ \text{KL}\left( \widehat{\mathbf{y}}_i^{t+1}, \sum_{j \in \mathcal{N}_\omega(\mathbf{x}_i)} \alpha_{ij} \widehat{\mathbf{y}}_j^{t+1} \right) \right] \tag{5}$$

Here, $\mathcal{N}_\omega(\mathbf{x}_i)$ denotes the indices of the global k-nearest neighbors of instance $\mathbf{x}_i$ within $\boldsymbol{s}^{t+1}$, calculated using embeddings extracted by $f_\omega$. The attention weight $\alpha_{ij}$ is computed in a similar manner as in Eq. (2), but based on the feature extractor $f_\omega$. This reward function evaluates the statistical smoothness of the labels based on local data structures from a prediction perspective, aligning with the ultimate goal of supporting prediction model training.

**Noisy Label Alignment Reward**   In addition, to evaluate the stability and robustness of the label correction action $\boldsymbol{a}^t$, we divide the data in state $\boldsymbol{s}^{t+1}$ into two subsets: a *clean* subset $\mathcal{D}_{\text{cle}}^{t+1}$ that contains the indices of instances whose labels are not changed—i.e., $a_i^t = 0$, and a *noisy* subset $\mathcal{D}_{\text{noi}}^{t+1}$ that contains the indices of instances whose labels are corrected—i.e., $a_i^t = 1$. We design the noisy label alignment (NLA) reward function, $\mathcal{R}_{\text{NLA}}$, to assess how well the *noisy* labels in $\mathcal{D}_{\text{noi}}^{t+1}$ align with the *clean* labels in $\mathcal{D}_{\text{cle}}^{t+1}$ based on an inter-subset k-nearest neighbor label prediction mechanism, such that:

$$\mathcal{R}_{\text{NLA}}(\boldsymbol{s}^t, \boldsymbol{a}^t) = -\mathbb{E}_{i \in \mathcal{D}_{\text{noi}}^{t+1}}\left[\text{KL}\left(\widehat{\mathbf{y}}_i^{t+1}, \sum_{j \in \mathcal{N}_{\text{cle}}(\mathbf{x}_i)} \alpha_{ij}\widehat{\mathbf{y}}_j^{t+1}\right)\right] \tag{6}$$

Here, $\mathcal{N}_{\text{cle}}(\mathbf{x}_i)$ represents the k-nearest neighbors identified in the clean subset for an instance $\mathbf{x}_i$ from the noisy subset. Both $\mathcal{N}_{\text{cle}}(\mathbf{x}_i)$ and the attention weights $\{\alpha_{ij}\}$ are again computed using embeddings extracted by $f_\omega$. The alignment—negative KL-divergence—between each *noisy* label $\widehat{\mathbf{y}}_i^{t+1}$ and the attention-based aggregation from its k-nearest *clean* neighbors reflects the statistical consistency of the label correction applied on $\mathbf{x}_i$ by the action $\boldsymbol{a}^t$.

The composite reward function $\mathcal{R}(\boldsymbol{s}^t, \boldsymbol{a}^t)$ integrates the two sub-reward functions introduced above as follows:

$$\mathcal{R}(\boldsymbol{s}^t, \boldsymbol{a}^t) = \exp\left(\mathcal{R}_{\text{LCR}}(\boldsymbol{s}^t, \boldsymbol{a}^t) + \lambda\mathcal{R}_{\text{NLA}}(\boldsymbol{s}^t, \boldsymbol{a}^t)\right) \tag{7}$$

where $\lambda$ is a trade-off hyper-parameter that balances the contributions of the two sub-reward functions. Since the value of negative KL-divergence is non-positive and unbounded, we deploy the exponential function, $\exp(\cdot)$, to rescale the reward values to the range of $(0, 1]$. This scaling mechanism is crucial for ensuring that the rewards remain bounded and normalized, facilitating a stable learning process.

### 3.2.4   Actor-Critic Method

Based on the RL formulation of the noisy label correction problem, defined through the key MDP components above, we adopt an actor-critic framework to learn the policy function by maximizing the expected cumulative reward.

In this framework, the policy function $\pi_\theta$ is considered as an "actor", while an action-value function $Q_\phi(\boldsymbol{s}, \boldsymbol{a})$ parameterized by $\phi$ is introduced as a "critic" to directly estimate the Q-value, defined as $Q(\boldsymbol{s}, \boldsymbol{a}) = \mathbb{E}_{\pi_\theta}[\sum_{t=0}^{\infty} \gamma^t \mathcal{R}(\boldsymbol{s}^t, \boldsymbol{a}^t) | \boldsymbol{s}^0 = \boldsymbol{s}, \boldsymbol{a}^0 = \boldsymbol{a}]$, which represents the expected discounted cumulative reward from taking action $\boldsymbol{a}$ in state $\boldsymbol{s}$. The learning objective for the policy function in the actor-critic method can be written as:

$$J(\pi_\theta) = \mathbb{E}_{\boldsymbol{s} \sim \rho_{\pi_\theta}}\left[\sum_{\boldsymbol{a}} \pi_\theta(\boldsymbol{a}|\boldsymbol{s})Q(\boldsymbol{s}, \boldsymbol{a})\right] \tag{8}$$

where $\rho_{\pi_\theta}$ denotes the stationary state distribution. The gradient of the objective over $\theta$ can be expressed as:

$$\nabla_\theta J(\pi_\theta) = \mathbb{E}_{\boldsymbol{s} \sim \rho_{\pi_\theta}, \boldsymbol{a} \sim \pi_\theta(\boldsymbol{s})}\left[\nabla_\theta \log \pi_\theta(\boldsymbol{a}|\boldsymbol{s})Q(\boldsymbol{s}, \boldsymbol{a})\right]. \tag{9}$$

The Q-values involved in the objective above are estimated using the critic network $Q_\phi$. To learn the critic network, we use the SARSA method to compute the Temporal Difference (TD) error [42, 26] for making on-policy updates to the critic function. The TD error at time-step $t - 1$ is given by:

$$\delta^{t-1} = \mathcal{R}(\boldsymbol{s}^{t-1}, \boldsymbol{a}^{t-1}) + \gamma Q(\boldsymbol{s}^t, \boldsymbol{a}^t) - Q_\phi(\boldsymbol{s}^{t-1}, \boldsymbol{a}^{t-1}) \tag{10}$$

The parameters of the critic network, $\phi$, are updated via the following stochastic gradient step using the TD error $\delta^{t-1}$:

$$\phi \leftarrow \phi + \beta\delta^{t-1}\nabla_\phi Q_\phi(\boldsymbol{s}^{t-1}, \boldsymbol{a}^{t-1}) \tag{11}$$

where $\beta$ is the learning rate for the critic.

**Input Encoding for the Critic**  The critic network, $Q_\phi$, requires both the state and action as inputs. Given the potentially large dimensions of the states and actions corresponding to the size of the training dataset, an efficient input encoding scheme is essential for reducing the computational cost. To this end, we devise a simple yet effective two step encoding scheme. First, given that our proposed RLNLC framework utilizes a deterministic transition mechanism, the next state $s^{t+1}$ is uniquely determined by the current state-action pair $(s^t, a^t)$. Consequently, we can use $s^{t+1}$ to replace the corresponding input pair $(s^t, a^t)$. We calculate $r(\mathbf{x}_i, \widehat{\mathbf{y}}_i^{t+1})$ as reward for a single pair of instance $(\mathbf{x}_i, \widehat{\mathbf{y}}_i^{t+1})$ with the same principle as Label Consistency Reward:

$$r(\mathbf{x}_i, \widehat{\mathbf{y}}_i^{t+1}) = \exp\left(- \text{KL}\Big(\widehat{\mathbf{y}}_i^{t+1}, \sum_{j \in \mathcal{N}_\omega(\mathbf{x}_i)} \alpha_{ij} \widehat{\mathbf{y}}_j^{t+1}\Big)\right) \tag{12}$$

where $\mathcal{N}_\omega(\mathbf{x}_i)$ is the indices of neighbor set of $f_\omega(\mathbf{x}_i)$ in $\mathcal{D}$, and $\{\alpha_{ij}\}$ are calculated using Eq. (2). The $\exp(.)$ function is used to normalize the reward and bound it in $(0, 1]$. Next, we employ a binning strategy to encode the state $s^{t+1} = \{(\mathbf{x}_i, \widehat{\mathbf{y}}_i^{t+1})\}_{i=1}^N$ based on the reward evaluations, substantially reduces its dimensionality. Specifically, we consider $N_b$ ($N_b \ll N$) number of bins and allocate each instance-label pair $(\mathbf{x}_i, \widehat{\mathbf{y}}_i^{t+1})$ in $s^{t+1}$ to one bin based on the following rule:

$$(\mathbf{x}_i, \widehat{\mathbf{y}}_i^{t+1}) \in \mathcal{B}_j \quad \text{if } r(\mathbf{x}_i, \widehat{\mathbf{y}}_i^{t+1}) \in \left(\frac{j-1}{N_b}, \frac{j}{N_b}\right], \tag{13}$$

where $\mathcal{B}_j$ denotes the $j$-th bin with $j \in [1 : N_b]$, and $r(\mathbf{x}_i, \widehat{\mathbf{y}}_i^{t+1})$ is the exponential of label consistency reward computed on the single given instance using Eq. (12). After binning, we construct a vector $\mathbf{v}^{t+1}$ with length $N_b$ to encode the state $s^{t+1}$, with each entry representing the proportion of instances allocated to the corresponding bin, such that:

$$\mathbf{v}_j^{t+1} = |\mathcal{B}_j|/N \tag{14}$$

where $|\mathcal{B}_j|$ denotes the number of instances in the $j$-th bin. This encoding process is deployed as part of the critic network $Q_\phi$ to transform the inputs $(s^t, a^t)$ into a simple vector $\mathbf{v}^{t+1}$, facilitating subsequent learning and promoting computational efficiency.

### 3.3  Label Cleaning for Prediction Model Training

After learning the policy function $\pi_\theta$ using the actor-critic method, we deploy the trained policy function for $T'$ time-steps to perform label cleaning on the noisy training dataset $\mathcal{D}$. This creates a trajectory with length $T'$ to progressively correct the noisy labels, starting from the initial state $s_0^0$. The labels obtained in the last state $s^{T'} = \{(\mathbf{x}_i, \widehat{\mathbf{y}}_i^{T'})\}_{i=1}^N$ are treated as "cleaned" labels. To promote efficiency, the prediction model $h_\psi \circ f_\theta$ is pre-trained on the given dataset $\mathcal{D}$ with noisy labels using standard cross-entropy loss. The feature extraction network $f_\theta$ then is further trained as part of the policy function learning. The final prediction model $h_\psi \circ f_\theta$ is obtained by further fine-tuning it on the "cleaned" data in $s^{T'}$ with the standard cross-entropy loss. The learning process of the proposed RLNLC method is summarized in Algorithm 1.

## 4  Experiments

### 4.1  Experimental Setup

**Datasets**  In the experiments, we employ four benchmark datasets to evaluate RLNLC under diverse noise conditions. CIFAR10-IDN and CIFAR100-IDN, adapted from CIFAR datasets [46], contain 50,000 training and 10,000 test images across 10 and 100 classes, respectively. Following prior work [7], instance-dependent label noise is injected into their training sets to mimic realistic corruption. Animal-10N [47] includes ten visually similar animal classes with 50,000 training and 5,000 test images, featuring roughly 8% noisy labels. Food-101N [22] consists of 310,009 web-sourced images over 101 categories with about 20% label noise, evaluated using the clean Food-101 test split of 25,250 manually verified images. Together, these datasets encompass experimental scenarios with different levels of label noise complexity.

---

**Algorithm 1** RLNLC Training Algorithm

---

**Input:** Initialized policy network $\pi_\theta$ and critic network $Q_\phi$; static initial state $\boldsymbol{s}_0^0$.
**Output**: Trained policy network parameters $\theta$ and critic network parameters $\phi$.
**for** each epoch **do**
   Randomly modify some labels in $\boldsymbol{s}_0^0$ to create the initial state $\boldsymbol{s}^0$.
   **for** $t = 0, 1, 2, \ldots, T$ **do**
      Form $\{\mathcal{N}(\mathbf{x}_i)\}_{i=1}^N$ for all $\mathbf{x}_i \in \mathcal{D}$.
      Compute $\{\alpha_{ij}\}_{i=1}^N$ for $j \in \mathcal{N}(\mathbf{x}_i)$ using Eq. (2).
      Compute the predicted labels $\{\bar{\mathbf{y}}_i\}_{i=1}^N$ using Eq. (1).
      Sample an action $\boldsymbol{a}^t \sim \pi_\theta(\cdot|\boldsymbol{s}^t)$ based on the probabilities computed using Eq. (3).
      Transition to the next state $\boldsymbol{s}^{t+1}$ by taking action $\boldsymbol{a}^t$ using Eq. (4).
      Compute $\mathcal{R}_{\mathrm{LCR}}$ and $\mathcal{R}_{\mathrm{NLA}}$ and combine them to store reward $\mathcal{R}(\boldsymbol{s}^t, \boldsymbol{a}^t)$ using Eq. (5), Eq. (6), and Eq. (7) .
      Compute and store Q-value: $Q(\boldsymbol{s}^t, \boldsymbol{a}^t) = Q_\phi(\boldsymbol{s}^t, \boldsymbol{a}^t)$.
      $\theta \leftarrow \theta + \beta_\theta \nabla_\theta \log \pi_\theta(\boldsymbol{a}^t|\boldsymbol{s}^t) Q(\boldsymbol{s}^t, \boldsymbol{a}^t)$.
      **if** $t \geq 1$ **then**
         $\delta^{t-1} = \mathcal{R}(\boldsymbol{s}^{t-1}, \boldsymbol{a}^{t-1}) + \gamma Q(\boldsymbol{s}^t, \boldsymbol{a}^t) - Q_\phi(\boldsymbol{s}^{t-1}, \boldsymbol{a}^{t-1})$.
         $\phi \leftarrow \phi + \beta \delta^{t-1} \nabla_\phi Q_\phi(\boldsymbol{s}^{t-1}, \boldsymbol{a}^{t-1})$.
      **end if**
   **end for**
**end for**

---

Table 1: Test accuracy (%) of different methods on CIFAR10-IDN and CIFAR100-IDN under various IDN noise rates. Standard deviations are shown as subscripts in parentheses. Columns correspond to different label noise ratios. $^\dagger$ denotes results reproduced using publicly available source code.

| Method | CIFAR10-IDN | | | | | CIFAR100-IDN | | | | |
|---|---|---|---|---|---|---|---|---|---|---|
| | 0.20 | 0.30 | 0.40 | 0.45 | 0.50 | 0.20 | 0.30 | 0.40 | 0.45 | 0.50 |
| CE [6] | 75.8 | 69.2 | 62.5 | 51.7 | 39.4 | 30.4 | 24.2 | 21.5 | 15.2 | 14.4 |
| Mixup [43] | 73.2 | 72.0 | 61.6 | 56.5 | 49.0 | 32.9 | 29.8 | 25.9 | 23.1 | 21.3 |
| Forward [44] | 74.6 | 69.8 | 60.2 | 48.8 | 46.3 | 36.4 | 33.2 | 26.8 | 21.9 | 19.3 |
| Reweight [19] | 76.2 | 70.1 | 62.6 | 51.5 | 45.5 | 36.7 | 31.9 | 28.4 | 24.1 | 20.2 |
| Decoupling [20] | 78.7 | 75.2 | 61.7 | 58.6 | 50.4 | 36.5 | 30.9 | 27.9 | 23.8 | 19.6 |
| Co-teaching [11] | 81.0 | 78.6 | 73.4 | 71.6 | 45.9 | 38.0 | 33.4 | 28.0 | 25.6 | 24.0 |
| MentorNet [10] | 81.0 | 77.2 | 71.8 | 66.2 | 47.9 | 38.9 | 34.2 | 31.9 | 27.5 | 24.2 |
| DivideMix [17] | 94.8 | 94.6 | 94.5 | 94.1 | 93.0 | 77.1 | 76.3 | 70.8 | 57.8 | 58.6 |
| CausalNL [6] | 81.4 | 80.3 | 77.3 | 78.6 | 67.3 | 41.4 | 40.9 | 34.0 | 33.3 | 32.1 |
| SSR$^\dagger$ [24] | 96.5 | 96.5 | 96.3 | 95.9 | 94.1 | 78.8 | 78.6 | 77.0 | 75.0 | 72.8 |
| RLNLC (Ours) | $\mathbf{97.3}_{(0.1)}$ | $\mathbf{97.1}_{(0.1)}$ | $\mathbf{96.9}_{(0.2)}$ | $\mathbf{96.6}_{(0.2)}$ | $\mathbf{95.8}_{(0.4)}$ | $\mathbf{80.5}_{(0.7)}$ | $\mathbf{80.1}_{(0.7)}$ | $\mathbf{78.5}_{(0.8)}$ | $\mathbf{77.2}_{(0.8)}$ | $\mathbf{74.7}_{(0.9)}$ |

**Implementation Details** In line with previous research [18, 48], our experiments employed a ResNet-34 backbone for CIFAR10-IDN and a ResNet-50 for CIFAR100-IDN and Food-101N. For Animal-10N, a VGG-19 with batch normalization, as outlined in prior work [47], is utilized. We also used a ResNet-18 for experiments with symmetric noise on CIFAR10 and CIFAR100. Our training methodology involved stochastic gradient descent (SGD) with a momentum of 0.9 and a batch size of 128. We also implemented L2 regularization with a coefficient of $5 \times 10^{-4}$. We started with an initial learning rate of 0.01, which was reduced to 0.001 at the halfway point of the training duration. Additionally, a preliminary warmup phase was conducted for the first 50 epochs. We train the policy network on the CIFAR10-IDN, CIFAR100-IDN, and Animal-10N datasets for 500 epochs and set $T = 10$. Subsequently, we evaluate the trained policy on the training data with a trajectory length of $T' = 25$. Using the corrected labels, we fine-tune the prediction model for an additional 100 epochs. The parameters $\lambda, \tau, \gamma, N_b$, and $k$ are set to 0.5, 0.5, 0.9, 100, and 10, respectively.

## 4.2 Comparison Results

We compare the proposed RLNLC with a set of methods, including CE [6], Mixup [43], Forward [44], Reweight [19], Decoupling [20], Co-teaching [11], Co-teaching+ [12], MentorNet [10], DivideMix [17], CausalNL [6], SSR [25], SSR+ [25], Nested-Dropout [21], CE+Dropout [21], SELFIE [47], PLC [15], Nested-CE [21], CleanNet [22], BARE [23], DeepSelf [45], PLC [15], LongReMix [18], and SURE [25]. We record the average results of RLNLC over five independent runs.

Table 2: Test accuracy (%, mean and standard deviation) on Animal-10N. † denotes results reproduced using publicly available source code.

| Method | CE | N-Dropout | CE+Dropout | SELFIE | PLC | Nested-CE | SSR† | SSR+ | SURE | RLNLC (Ours) |
|---|---|---|---|---|---|---|---|---|---|---|
| | $79.4_{(0.1)}$ | 81.8 | $81.3_{(0.3)}$ | $81.8_{(0.1)}$ | $83.4_{(0.4)}$ | $84.1_{(0.1)}$ | 87.7 | 88.5 | 89.0 | $\mathbf{90.2}_{\mathbf{(0.1)}}$ |

Table 3: Test accuracy (%, mean and standard deviation) on Food-101N.

| Method | CE[6] | CleanNet[22] | BARE[23] | DeepSelf[45] | PLC[15] | LongReMix[18] | RLNLC (Ours) |
|---|---|---|---|---|---|---|---|
| | 81.6 | 83.9 | 84.1 | 85.1 | $85.2_{(0.0)}$ | 87.3 | $\mathbf{89.2}_{\mathbf{(0.1)}}$ |

Table 1 presents the performance of our method compared to existing techniques with various noise rates on the CIFAR10-IDN and CIFAR100-IDN datasets, utilizing ResNet-34 and ResNet-50 as backbone networks, respectively. Our proposed method, RLNLC, consistently produces the best results. On CIFAR10-IDN, RLNLC achieves a noteworthy improvement of 1.7% over the second best method, SSR, with a large noise rate of 0.50, highlighting our method's ability to maintain high accuracy under elevated noise conditions. On CIFAR100-IDN, our RLNLC consistently outperforms the nearest competitor, SSR, by margins of 1.7%, 1.5%, 1.5%, 2.2%, and 1.9% across noise rates of 0.20, 0.30, 0.40, 0.45, and 0.50, respectively. This sustained superiority across various noise intensities demonstrates the robustness and adaptability of our approach, affirming its effectiveness in handling complex noisy label scenarios.

The results in Table 2 showcase the comparative performance of our proposed RLNLC on the Animal-10N dataset, with VGG-19 as the backbone network. Our method achieves a leading test accuracy of 90.2%, marking a significant improvement of 1.2% over the next best method, SURE. This enhancement underscores the advantage our approach provides over established techniques.

Table 3 presents the comparative results on the Food-101N dataset employing ResNet-50 as the backbone network. RLNLC achieves the best test accuracy of 89.2% , outperforming all competing methods by a significant margin. Specifically, we observe an improvement of approximately 1.9% over the second best-performing method, LongReMix.

We also conducted experiments on CIFAR10 and CIFAR100 with symmetric noise [44], a type of class-conditioned noise, using ResNet-18 as the backbone network. The comparison results across different noise rates are reported in Table 4. Our RLNLC exhibits superior performance on both CIFAR10 and CIFAR100 under high symmetric noise conditions. Notably, at a 90% noise level on CIFAR100, RLNLC achieved an impressive test accuracy of 44.2%, significantly surpassing DivideMix, which recorded only 31%. This 13.2 percentage point increase emphasizes the robustness and effectiveness of RLNLC in handling extreme noisy scenarios. Similarly, on CIFAR10 with the same noise level, RLNLC reached an accuracy of 82.1%, outperforming DivideMix by 6.7 percentage points. Such quantitative enhancements validate the superior capabilities of our method in maintaining accuracy despite high levels of label noise.

## 4.3 Ablation Study

We conducted an ablation study to investigate the impact of different components in our proposed RLNLC framework on the overall performance. The study was carried out on the CIFAR100-IDN dataset, with noise rates ranging from 0.20 to 0.50. Specifically, we considered four ablation variants: (1) "$-$w/o $\mathcal{R}_{\mathrm{NLA}}$" removes the noisy label alignment reward; (2) "$-$w/o $\mathcal{R}_{\mathrm{LCR}}$" drops the label consistency reward; (3) "$-$w/o random. $s^0$" drops initial state randomization and starts each epoch's training from the fixed initial state $s^0 = s_0^0 = \mathcal{D}$; (4) "$f_\omega \leftarrow f_\theta$" uses the policy network feature extractor for reward evaluation. The comparison results are reported in Table 5. We can see that removing any key component from RLNLC leads to a noticeable decrease in performance across all noise levels. Specifically, the removal of $\mathcal{R}_{\mathrm{NLA}}$ results in lower accuracy, highlighting its crucial role in leveraging clean data to guide the correction of noisy labels within the dataset. Similarly, excluding the label consistency reward, $\mathcal{R}_{\mathrm{LCR}}$, diminishes the system's ability to maintain local label consistency, which is essential for the model's robust performance across various noise conditions. The variant without initial state randomization shows a smaller decline in performance compared to the first two ablated conditions but still underperforms relative to the full RLNLC model. This suggests that slight initial state randomization helps the model in exploring more diverse corrective strategies. Finally, the variant "$f_\omega \leftarrow f_\theta$" leads to a clear performance drop, demonstrating that decoupling the policy and reward networks is essential for stable reinforcement learning.

Table 4: Test accuracy (%) of different methods on CIFAR10 and CIFAR100 under various symmetric noise rates. Columns correspond to different label noise ratios. Bold values indicate the best results.

| Method | CIFAR10 | | | CIFAR100 | | |
|---|---|---|---|---|---|---|
| | 0.50 | 0.80 | 0.90 | 0.50 | 0.80 | 0.90 |
| CE [17] | 57.9 | 26.1 | 16.8 | 37.3 | 8.8 | 3.5 |
| Co-teaching+ [12] | 84.1 | 45.5 | 30.1 | 45.3 | 15.5 | 8.8 |
| Mixup [43] | 77.6 | 46.7 | 43.9 | 46.6 | 17.6 | 8.1 |
| DivideMix [17] | 94.4 | 92.9 | 75.4 | 74.2 | 59.6 | 31.0 |
| RLNLC (Ours) | $\mathbf{97.4}_{(0.1)}$ | $\mathbf{95.8}_{(0.2)}$ | $\mathbf{82.1}_{(0.7)}$ | $\mathbf{81.2}_{(0.3)}$ | $\mathbf{70.6}_{(0.4)}$ | $\mathbf{44.2}_{(0.8)}$ |

Table 5: Results of ablation study in terms of test accuracy on CIFAR100-IDN . Top row presents the label noise ratios. Bold font indicates the best results.

| Method | 0.20 | 0.30 | 0.40 | 0.45 | 0.50 |
|---|---|---|---|---|---|
| RLNLC (Ours) | $\mathbf{80.5}_{(0.7)}$ | $\mathbf{80.1}_{(0.7)}$ | $\mathbf{78.5}_{(0.8)}$ | $\mathbf{77.2}_{(0.8)}$ | $\mathbf{74.7}_{(0.9)}$ |
| $-$w/o $\mathcal{R}_{\mathrm{NLA}}$ | $78.4_{(0.4)}$ | $77.9_{(0.8)}$ | $76.2_{(0.6)}$ | $76.3_{(0.8)}$ | $72.0_{(0.9)}$ |
| $-$w/o $\mathcal{R}_{\mathrm{LCR}}$ | $79.3_{(0.6)}$ | $78.5_{(0.7)}$ | $76.1_{(0.9)}$ | $76.1_{(0.6)}$ | $73.9_{(0.8)}$ |
| $-$w/o random. $\boldsymbol{s}^0$ | $79.9_{(0.6)}$ | $79.5_{(0.9)}$ | $77.8_{(0.5)}$ | $76.8_{(0.9)}$ | $73.1_{(0.9)}$ |
| $f_\omega \leftarrow f_\theta$ | $78.4_{(0.8)}$ | $76.9_{(0.8)}$ | $75.2_{(0.6)}$ | $74.3_{(0.8)}$ | $73.8_{(0.9)}$ |

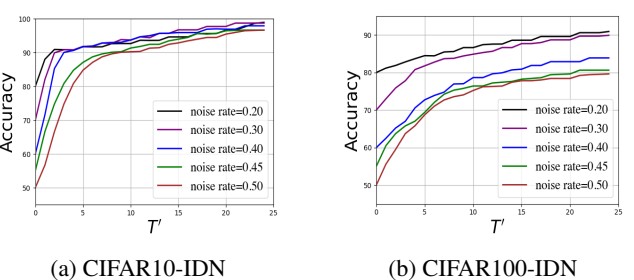

(a) CIFAR10-IDN  (b) CIFAR100-IDN

Figure 2: Label correction accuracy on the training set by deploying the trained policy function for $T'$ time-steps. Results on CIFAR10-IDN and CIFAR100-IDN with various noise rates are plotted.

## 4.4 Label Correction Accuracy

We tested the label correction accuracy on CIFAR10-IDN and CIFAR100-IDN, examining how varying noise rates in $\{0.2, 0.3, 0.4, 0.45, 0.5\}$ impacts the label correction performance during the $T'$ noise correction steps on the training set using the trained policy function. The results are presented in Figure 2. The CIFAR10-IDN dataset reveals a clear distinction in label correction accuracy improvement across different noise rates; lower noise levels (0.2, 0.3, 0.4) exhibit rapid convergence, achieving over 90% accuracy by $T' = 5$. Even with the highest noise rate (0.50) our method yields 90% accuracy by $T' = 10$, indicating our approach's robustness to high noise rates. On the more complex CIFAR100-IDN dataset, our label correction strategy adeptly handles the increased class count and intrinsic dataset complexities, albeit with slightly lower accuracies. This is consistent with expectations given the heightened difficulty of the task. Achieving around 90% accuracy in lower noise conditions and maintaining over 79% accuracy even in high noise settings on CIFAR100-IDN showcases RLNLC's adaptability.

## 5 Conclusion

In this paper, we innovatively formulated noisy label correction as a reinforcement learning problem and developed a novel RL approach, named as RLNLC, to address the problem of learning with noisy labels. RLNLC integrates stochastic policy-driven actions for label correction with a carefully crafted reward function that fosters both label consistency and noisy label alignment. Its capability to make sequential, non-myopic decisions is well-suited for handling label noise in complex environments, thereby enhancing the robustness of label corrections. Experiments are conducted on multiple benchmark datasets, and the experimental results demonstrated that RLNLC consistently outperforms existing state-of-the-art methods. These findings underscore the potential of reinforcement learning to transform the landscape of learning in noisy environments.

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

Figure 3: Sensitivity analysis for four hyper-parameters, $k$, $N_b$, $\lambda$, and $T$, on CIFAR100-IDN with 0.50 noise rate.

## A  Hyper-parameter Sensitivity Analysis

We conducted a detailed sensitivity analysis of four key hyperparameters within the RLNLC framework on CIFAR100-IDN: (1) $k$—hyperparameter for number of neighbors in k-nearest neighbors, (2) $N_b$—hyperparameter for the number of bins for state encoding, (3) $\lambda$—trade-off coefficient for $\mathcal{R}_{\mathrm{NLA}}$, and (4) $T$—trajectory length in the training procedure, revealing their influence on the performance. The experimental results for various hyper-parameter values are plotted in Figure 3.

We can see that the test accuracy performance improves as $k$ increases from 3 to 10, highlighting the benefits of a broader contextual basis for label refinement. However, further increases beyond 10 yield diminishing returns, with accuracy plateauing around 74.9% for $k$ values of 20 and 30. This suggests that while a larger contextual window is beneficial up to a point, excessively wide windows offer no extra benefits, likely due to less informative examples. The granularity of state encoding, controlled by $N_b$, shows a similar trend where increasing $N_b$ from 10 to 100 leads to improved performance. Further increases up to 500 continue to marginally enhance the model's accuracy. This improvement shows that finer state space discretization enables more precise state encoding, though gains diminish with increasing $N_b$. The optimal range for $\lambda$ is between 0.4 and 0.6, achieving a peak accuracy of 74.9%. Outside this range, effectiveness drops, indicating an imbalance that may reduce learning efficiency. Varying the training trajectory length $T$ shows an initial increase in performance as $T$ increases from 1 to 10. However, extending $T$ further to 25 leads to a decline in performance. This reduction may stem from overfitting to noisy data or excessive refinement causing label drift.

## B  Computer resources

Our experiments were performed using computing systems equipped with 8-core Intel Core processors, 64 GB of system memory, and NVIDIA GeForce RTX 3060 GPUs, each providing 12 GB of dedicated video memory.

## C  Limitation

RLNLC targets the standard learning-with-noisy-labels setting under a single, stationary data distribution, without explicitly modeling domain shift or class imbalance. Nonetheless, the proposed formulation is general and flexible enough to be extended to more complex scenarios. These extensions would build upon the core principles of RLNLC, and thus represent natural and promising directions for future work.

## D  Broader Impacts

This work proposes a reinforcement learning-based framework for learning with noisy labels, which can improve the reliability and robustness of machine learning models trained on imperfect datasets. By reducing the reliance on clean labels, RLNLC can make it more feasible to leverage large-scale datasets in domains where annotation is expensive or error-prone, such as medical imaging, remote sensing, or crowd-sourced labeling. However, automatic label correction methods also introduce potential risks, such as reinforcing existing biases in the data or misclassifying minority or rare samples if not carefully validated. Future deployments should therefore include safeguards, such as human-in-the-loop verification or fairness audits, to ensure responsible use.

