# OpenReview forum: "Learning to Clean: Reinforcement Learning for Noisy Label Correction"
_NeurIPS.cc/2025/Conference — NeurIPS 2025 poster_

### Official Review · Reviewer_un5g · 2025-06-30

**Clarity:** 3
**Significance:** 3
**Originality:** 3
**Rating:** 5
**Confidence:** 1

**Summary:**

This paper proposes a novel framework for addressing the problem of label noise by formulating label correction as a reinforcement learning problem. The authors introduce the RLNLC method, which constructs a Markov Decision Process (MDP) that includes a state space, action space, and reward mechanism. A policy network is trained using an actor-critic framework to correct potentially mislabeled data. For the reward function, the authors design two submodules—label consistency and clean-label alignment—to guide the policy network in generating effective label correction sequences. Experiments conducted on multiple benchmark datasets demonstrate the strong generalization ability and robustness of RLNLC.

**Questions:**

1. Noted that the k-means method is used in the reward function. During training state, there may be cases where the number of samples with certain labels in a batch is fewer than k. In such situations, the computed reward may deviate from the intended objective, potentially leading to suboptimal guidance for the policy network. Will this method be affected by this situation?

2. Could the repeated correction of labels introduce new errors or cause the model to diverge from plausible semantics (especially in datasets with inherent label ambiguity)?

3. Can the authors provide a comparison of the number of parameters and computational cost between RLNLC and other baseline methods?

**Ethical Concerns:**

["NO or VERY MINOR ethics concerns only"]

**Final Justification:**

For the first issue, the authors offered a clear and thorough explanation regarding the effect of the hyper parameter k on the model, including how different values of k impact the reward function. This resolved my main concern.

For the second issue, the authors responded from multiple perspectives and noted that no such problem was observed in their experiments. This helped alleviate the concern I raised.

For the third issue, the authors responded directly with runtime comparisons against different baseline methods, highlighting the efficiency of their approach.

My concerns have been adequately addressed. Since the first issue was my primary focus, and the authors’ response was meaningful and improved my perception of the paper’s clarity, I have reassessed the manuscript. Additionally, the response to the third issue strengthened my view of the paper’s significance by demonstrating a clear advantage in computational efficiency.

Taking all aspects into account, I have adjusted my score accordingly.

**Limitations:**

yes

**Paper Formatting Concerns:**

No Formatting concern.

**Quality:**

3

**Strengths And Weaknesses:**

Strengths
1. The idea of casting noisy label correction as a reinforcement learning problem is both original and compelling.
2. The MDP elements states, actions, and rewards are clearly defined and grounded in label consistency principles. Key components such as the reward terms and randomized initialization are empirically validated.
Weaknesses
1. While the method is empirically effective, there is little theoretical analysis of convergence properties, stability of the policy under label noise, or sample complexity in the RL setting.
2. The actor-critic framework, kNN-based reward computation, and multiple label correction steps may introduce significant computational costs.

---

> ### Author Rebuttal · Authors · 2025-07-29
>
> We sincerely appreciate the time and effort the reviewer has dedicated to reviewing our work.
>
> **About insufficient samples**: We clarify that the reward function uses k-NN, not k-means. Additionally, the state representation in RLNLC is built from the **entire dataset**, not just a mini-batch, so k-NN is computed globally over all feature embeddings. This ensures that each sample has access to a sufficiently large context, thereby avoiding issues arising from limited local information.
>
>
> **About the effect of repeated corrections**:RLNLC mitigates this risk of diversion through several design choices. First, label corrections are guided by a stochastic policy trained to maximize long-term reward, which explicitly encodes both local consistency and alignment with confidently clean labels via k-NN mechanisms. This ensures that updates are not arbitrary but grounded in dataset-wide structure. Second, corrections are applied softly and probabilistically, avoiding hard deterministic overwrites that could amplify noise. Finally, the actor-critic framework evaluates the downstream impact of corrections via reward feedback, discouraging degenerate or unstable policies that would introduce semantic drift. Empirically, we observe stable convergence and improved performance across diverse datasets, indicating the model does not diverge but instead learns semantically coherent label refinements.
>
> **About number of parameters**:  We emphasize that RLNLC uses the exact same backbone architecture as all baselines, ensuring a fair comparison in terms of model capacity. The only additional component is a lightweight critic network composed of a few MLP layers, which introduces roughly 3K additional trainable parameters, negligible relative to modern backbones such as ResNet, which has millions of parameters. In contrast, several widely used LNL baselines introduce substantially more overhead. For example, Co-teaching [1] and DivideMix [2] both maintain two separate networks during training, effectively doubling the number of backbone parameters. Despite its minimal overhead, RLNLC delivers significantly stronger results, showing that performance gains stem not from model size but from the structured, reward-driven label correction process.
>
> **About training time**: Our method (RLNLC) has a training time lower than most SOTA methods. The table shows the training time in seconds per epoch on CIFAR100-IDN using an RTX 3060 GPU.
>
> | Method         | Time  |
> |----------------|-------|
> | SSR            | 51.37 |
> | DivideMix      | 119.91|
> | LongRemix      | 140.80|
> | RLNLC (Ours)   | 59.51 |
>
>
> [1] B. Han, Q. Yao, X. Yu, G. Niu, M. Xu, W. Hu, I. Tsang, and M. Sugiyama, “Co-teaching:
> Robust training of deep neural networks with extremely noisy labels,” NeurIPS, 2018.
>
> [2] J. Li, R. Socher, and S. C. Hoi, “Dividemix: Learning with noisy labels as semi-supervised learning”, ICLR, 2020

---

> > ### Comment · Reviewer_un5g · 2025-08-05
> > **About insufficient samples**
> >
> > We apologize for the ambiguity in our first comment, which may have led to a misunderstanding. What we intended to ask is: in Equations (5) and (6), each element computes the KL divergence with its k nearest neighbors, and the choice of k plays a critical role in the reward function.
> >
> > From our understanding, elements with similar labels are more likely to be clustered together. However, if k is too large, is there a risk that data points with different labels could be included in the neighborhood, thereby affecting the reliability of the reward function? (This concern is partially motivated by the experimental results in Section D: Hyper-parameter Sensitivity Analysis.)
> >
> > Furthermore, even if k is not large, what happens in cases where there are fewer than k elements with the same label? Would the reward function still work as intended in such scenarios?

---

> > > ### Author Response · Authors · 2025-08-06
> > >
> > > We sincerely appreciate the reviewer's thoughtful follow-up questions, which allow us to clarify important aspects of our method. Below, we address the concerns about **neighborhood size (k)** in our reward function:
> > >
> > > **About risk of large k in reward computation**:
> > > The softmax-normalized attention weights $\alpha\_{ij}$ assign higher importance to neighbors *closest in feature space*. Even if $k$ is large, irrelevant neighbors (e.g., those with divergent labels) receive near-zero weight. This effectively "downweights" outliers, ensuring the aggregated label $\bar{\mathbf{y}}_i$ (Eq. 1) remains dominated by semantically similar neighbors.
> > >  Our sensitivity analysis (Appendix D) shows RLNLC performs robustly for $k \in [5, 50]$. The performance stops improving for k> 20 as excessively wide windows offer no extra benefits due to smaller weights assigned to added outliers to $k$.
> > >
> > > **About handling fewer than $k$ same-label instances**:
> > > The reward function prioritizes *feature-space similarity* (via $f\_\omega$ embeddings), not label counts. Even if same-label neighbors are scarce, neighbors with *compatible features* (e.g., visually similar images) still contribute meaningfully to $\bar{\mathbf{y}}\_i$. The KL divergence (Eq. 5, 6) naturally handles low-confidence predictions: if neighbors disagree, $\mathcal{R}\_{\mathrm{LCR}}$ and ${\mathcal{R}\_{\mathrm{NLA}}}$ penalize inconsistency, guiding the policy to avoid overcorrection.   In sparse regions, the attention mechanism (Eq. 2) concentrates weight on the few reliable neighbors available, avoiding dilution by irrelevant points.

---

> > > > ### Comment · Reviewer_un5g · 2025-08-07
> > > >
> > > > Thank you for the clarification, I am satisfied with the explanation provided. Therefore, I have adjusted my score to reflect this clarification.

---

### Official Review · Reviewer_Kkod · 2025-07-02

**Clarity:** 4
**Significance:** 3
**Originality:** 3
**Rating:** 5
**Confidence:** 3

**Summary:**

The paper proposes a framework that learns a policy for noisy label correction via reinforcement learning, with comprehensive formulations of the problem setup, state space, action space, reward functions, and an actor-critic method that is used to learn the desired policy function. The proposed framework addresses several drawback of prior works. Extensive experiments are also conducted to show the effectiveness of it.

**Questions:**

1. Line 33. The paper claims that the prior works lack the ability to learn from long-term consequences. What are considered "lone-term consequences" in this context, and how does the proposed method in this paper addresses the problem?
2. Line 42-43. The policy function separates the training set into a "clean" subset and a "noisy" subset. Here by saying "clean" and "noisy", does it mean whether the labels in the subsets are clean or noisy by nature, or does it mean whether the labels in the subsets are "predicted to be clean or noisy by the reinforcement model"?
3. Does the proposed RLNLC applies only on IDN or both IDN and CCN?
4. From my understanding of the framework of RLNLC, $f_\theta$ is to be trained, while $f_\omega$ is fixed for computing the reward function? If so, then what guarantees the performance of the fixed model itself? What guarantees that the reward function computed using $f_\omega$ indeed reflects the true "goodness" of the corrected labels?
5. Line 226. I suppose that by saying "trade-off" you are implying that if one of the sub-reward functions gets better, the other one will get worse? If so, can you explain this trade-off more concretely or intuitively?

**Ethical Concerns:**

["NO or VERY MINOR ethics concerns only"]

**Final Justification:**

Most of my questions have been addressed. I have maintained my original rating.

**Limitations:**

Yes.

**Quality:**

3

**Strengths And Weaknesses:**

Strengths:
1. Clear chain of logics.
2. The experiments are well conducted and sufficiently detailed。
3. The review on the related works is comprehensive.
4. The formulation of the proposed method is thorough and detailed.
Weaknesses:
1. The fixed backbone model itself doesn't have a performance guarantee.

---

> ### Author Rebuttal · Authors · 2025-07-29
>
> We sincerely appreciate the time and effort the reviewer has dedicated to reviewing our work.
>
> **About long-term consequences**: In the context of learning with noisy labels, "long-term consequences" refer to the downstream effects that early label correction decisions can have on the final performance of the prediction model, especially under instance-dependent noise, where errors can compound over time. Most prior works, such as Co-teaching [1] and DivideMix [2], rely on short-term heuristics (e.g., small-loss filtering) without accounting for how corrections influence future learning dynamics. In contrast, RLNLC frames label correction as a sequential decision-making problem using reinforcement learning. By modeling corrections as actions and using a reward function that evaluates dataset-wide label consistency and inter-subset alignment, RLNLC explicitly learns a policy that optimizes over future label quality and model behavior, not just immediate improvements. The actor-critic framework allows the system to assess whether a series of corrections improves the overall learning trajectory, thereby mitigating the risk of early missteps cascading into degraded final performance.
>
> **About clean and noisy in policy**: The terms "clean" and "noisy" refer to the policy’s predicted status of each sample’s label during training, i.e., whether the reinforcement learning policy believes the label is likely correct (clean) or incorrect (noisy). Since the true label correctness is unknown during learning, these designations reflect the policy’s internal belief rather than the ground truth. We will clarify this distinction more explicitly in the revision.
>
> **About noise type:** Our method is general and applies to class-conditional noise (CCN) as well as IDN. To demonstrate this, we include CCN results in Appendix C, Table 5, where RLNLC shows competitive performance, further confirming its robustness across different noise models.
>
> **About reward function network $f_\omega$**:  The role of $f_\omega$ is to provide stable embeddings for the reward function. Using a separate, fixed model avoids coupling the label correction policy with the reward signal, reducing potential bias. We emphasize three key points:
>
> 1. **Empirical Competence of $f_\omega$**:
>    Although trained on noisy labels, $f_\omega$ produces sufficiently informative representations, as reflected in its performance (CE rows in Tables 1–3 in our manuscript; for convince also included below). These features enable meaningful similarity estimation via k-NN, allowing the policy to reason about label quality based on local structure.
> Accuracy of $f_\omega$ (CE Baseline)
>
> | Dataset        | Noise Rate | Accuracy (%) |
> |----------------|------------|---------------|
> | CIFAR10-IDN    | 0.20       | 75.8          |
> |                | 0.30       | 69.2          |
> |                | 0.40       | 62.5          |
> |                | 0.45       | 51.7          |
> |                | 0.50       | 39.4          |
> | CIFAR100-IDN   | 0.20       | 30.4          |
> |                | 0.30       | 24.2          |
> |                | 0.40       | 21.5          |
> |                | 0.45       | 15.2          |
> |                | 0.50       | 14.4          |
> | ANIMAL-10N     | --         | 79.4          |
> | Food-101N      | --         | 81.6          |
>
>
>
> 2. **Policy Learning with Feedback**:
>    The policy is not blindly steered by $f_\omega$’s outputs. Instead, it learns through reinforcement feedback to calibrate its reliance on neighborhood signals. The reward captures both short- and long-term consistency, encouraging corrections that lead to better downstream fine-tuning. This dynamic loop allows RLNLC to iteratively refine labels beyond the limitations of the fixed model.
>
> 3. **Empirical Justification**:
>    The table below shows that using a separate $f_\omega$ for the reward yields better performance than reusing the policy model $f_\theta$. Results are reported as test accuracy (with standard deviation) on CIFAR-100 IDN.
>
>
> | Method             | 0.20           | 0.30          | 0.40          | 0.45              | 0.50              |
> |--------------------|----------------|---------------|---------------|-------------------|-------------------|
> | **RLNLC (Ours)**   | **80.5** (0.7) | **80.1** (0.7)| **78.5** (0.8)| **77.2** (0.8)    | **74.7** (0.9)    |
> | - w $f_\theta$   | 78.4 (0.4)     | 76.9 (0.8)    | 75.2 (0.6)    | 74.3 (0.8)        | 72.8 (0.9)        |
>
>
>
>
>
> **About trade-off**: We did not mean a zero-sum relationship between the reward terms. By “trade-off,” we refer to a **weighting hyperparameter** that balances the contributions of $\mathcal{R}\_{\text{NLA}}$ and $ \mathcal{R}\_{\text{LCR}}$ in the total reward. The effect of different values of $\lambda$ is illustrated in **Figure 3, Appendix D** of our manuscript. We will clarify this in the revision.
>
>
> [1] B. Han, Q. Yao, X. Yu, G. Niu, M. Xu, W. Hu, I. Tsang, and M. Sugiyama, “Co-teaching:
> Robust training of deep neural networks with extremely noisy labels,” NeurIPS, 2018.
>
> [2] J. Li, R. Socher, and S. C. Hoi, “Dividemix: Learning with noisy labels as semi-supervised learning”, ICLR, 2020

---

> ### Comment · Reviewer_Kkod · 2025-08-08
>
> Thank you for the detailed response. I was wondering if the fixed backbone model itself has a performance guarantee?

---

> > ### Author Response · Authors · 2025-08-09
> >
> > We appreciate the reviewer’s follow-up question.
> > We do not claim a formal theoretical guarantee on $f_\omega$  absolute classification accuracy, nor is such a guarantee typical in noisy-label learning, since the input data itself is corrupted.
> >
> > Instead, our design leverages **empirical sufficiency** and **relative stability**:
> >
> > 1. **Empirical Sufficiency** :
> >    Even when trained on noisy data, $f_\omega$ consistently produces embeddings with meaningful neighborhood structure. As shown in CE baseline rows in Tables 1–3, its standalone accuracy is competitive with standard noisy-label baselines, confirming that its features preserve class-discriminative geometry. The reward function relies on *relative neighborhood consistency*, not on perfectly correct predictions, making it robust to moderate representation noise.
> >
> > 2. **Relative Stability and Decoupling** :
> >    Freezing $f_\omega$ ensures that the reward signal remains stable throughout policy learning. If the backbone used for reward computation were updated jointly with the policy, the reward landscape would shift in tandem with the policy’s actions, introducing feedback loops that could bias or destabilize learning. The fixed model serves as a stable reference frame.
> >
> > 3. **Policy Feedback Loop**:
> >    The RLNLC policy is not blindly guided by $f_\omega$. Through reinforcement feedback, it learns to calibrate how much to trust the k-NN–based reward in different regions of the dataset. This adaptivity allows it to exploit the useful structure in $f_\omega$’s embeddings while compensating for any residual bias.
> >
> > 4. **Empirical Validation** :
> >    We experimentally compare RLNLC using a fixed $f_\omega$ vs. using the policy’s own evolving backbone for reward computation (see table in our rebuttal). The fixed-backbone version consistently yields better accuracy across noise levels, supporting the claim that stability, not perfect accuracy, is the key requirement.

---

### Official Review · Reviewer_FMfs · 2025-07-02

**Clarity:** 3
**Significance:** 2
**Originality:** 3
**Rating:** 4
**Confidence:** 5

**Summary:**

This paper introduces RLNLC, a novel reinforcement learning framework for noisy label correction, modeling the problem as a Markov Decision Process (MDP). By defining a state space of data and labels, an action space of label corrections, and a reward function combining label consistency and subset alignment, RLNLC uses an actor-critic method to learn a policy network for iterative label cleaning. Extensive experiments on benchmark datasets demonstrate its effectiveness.

**Questions:**

Please refer to the weaknesses.

**Ethical Concerns:**

["NO or VERY MINOR ethics concerns only"]

**Final Justification:**

The author's response adequately addresses the raised concerns, and their work, with its innovative RL-based label correction approach and well-designed components, shows promise, warranting a positive assessment.

**Limitations:**

Yes.

**Paper Formatting Concerns:**

None.

**Quality:**

3

**Strengths And Weaknesses:**

Strengths:
1. The RL-based approach to deal with label correction is a fresh idea, offering dynamic, sequential decision-making that traditional methods (e.g., sample selection or loss regularization) lack.

2. This work develops a customized policy function grounded in a deep representation network.

3. This work designs an effective reward function. Through k-nearest neighbor prediction mechanisms, it promotes efficient label correction by matching local data structures with reliable clean labels.


Weaknesses:
1. In Line 56, the paper states that its designed policy function determines label correction actions by considering the state of the entire dataset. However, as observed in Section 3.2.2, it appears to only consider data within a single batch or subset, which may constitute an overclaim. If my understanding is incorrect, please provide further clarification.

2. The paper does not compare with baselines under another noise setting (CCN). If such comparisons are omitted, it is necessary to add explanations, preferably highlighting that the algorithm is specifically designed for the more practical IDN noise setting.

3. In the paper, the figures need to be polished up. Figure 1 is too small and could be enlarged. The font in Figure 2 is slightly distorted, and the figure appears not to be in vector format. It is recommended to use vector graphics for a more professional look.

4. The iterative RL training and multi-step label correction may incur higher computational costs than non-iterative methods (e.g., CleanNet), though hardware details (e.g., training time on RTX 3060) are missing.

5. The related work section discusses and cites too few works.

---

> ### Author Rebuttal · Authors · 2025-07-29
>
> We sincerely appreciate the time and effort the reviewer has dedicated to reviewing our work.
>
> **About state**: The policy **does in fact operate on a global view of the dataset**. The k-NN aggregation in Section 3.2.2  is performed globally, leveraging the full feature space of the entire dataset at each policy step. This ensures that the correction probabilities incorporate the relative similarity of each sample to all others, not just the local batch context. Furthermore, to make the global structure tractable and informative for the policy network, we introduce a **state encoding mechanism (in Section 3.2.4)** that summarizes the dataset as a fixed-length representation via a binning strategy . This allows the policy to condition on a **compressed but global encoding of the full dataset state**, not just a partial view.
>
> **About CCN noise:** The comparison results under class-conditional noise (CCN) are provided in **Appendix C, Table 5** of our manuscript, demonstrating that RLNLC maintains strong performance even in this simpler noise setting.
>
> **About Figure**: We thank the reviewer for the helpful suggestion regarding the figure. We will incorporate the feedback and improve the figure in the revision.
>
> **About Related works**: We have discussed the most relevant and state-of-the-art works in learning with noisy labels, as well as the application of reinforcement learning in related problems. We appreciate the feedback and will include additional related works in the revision.
>
> **About training time**: Our method (RLNLC) has a training time lower than most SOTA methods. The table shows the training time in seconds per epoch on CIFAR100-IDN using an RTX 3060 GPU.
>
> | Method         | Time  |
> |----------------|-------|
> | SSR            | 51.37 |
> | DivideMix      | 119.91|
> | LongRemix      | 140.80|
> | RLNLC (Ours)   | 59.51 |
>
>
> .

---

> > ### Comment · Reviewer_FMfs · 2025-08-07
> >
> > The author's response has resolved most of my confusion. I hope the author can revise the article according to the suggestions in the final version, and I maintain a positive attitude towards this article.

---

### Official Review · Reviewer_uqZQ · 2025-07-03

**Clarity:** 3
**Significance:** 3
**Originality:** 3
**Rating:** 4
**Confidence:** 3

**Summary:**

This paper presents a framework for correcting noisy labels through a reinforcement learning approach. It incorporates two reward functions to assess the quality of actions: a label consistency reward function and a noisy label alignment reward function. Experimental results on four public datasets demonstrate the effectiveness of the proposed method, showing marginally better performance compared to other approaches.

**Questions:**

Please see my questions in the weakness section. My main concerns are the constraint on the noise levels and whether the proposed method can be used in multi-label classification tasks.

**Ethical Concerns:**

["NO or VERY MINOR ethics concerns only"]

**Final Justification:**

Based on the authors rebuttal, which solves my concern in this paper, I keep my original rating.

**Limitations:**

yes

**Quality:**

3

**Strengths And Weaknesses:**

Strengths:
1. Addressing the challenge of learning from noisy labels is a critical topic in the machine learning community. This paper introduces a novel approach by employing a reinforcement learning framework to correct noisy labels.

2. The paper proposes two informative reward functions that promote label correction by leveraging local data structures and clean labels.

3. Experimental results on four public datasets demonstrate that the proposed method outperforms the compared approaches.

Weaknesses:
1. More details are needed to understand Equation (3), specifically why it only considers the classes where the new prediction is equal to or greater than the original predicted class. What about cases where the new prediction has a smaller probability?

2. Regarding the feature extraction network ( f_\theta ), it is unclear which data was used to pretrain the model. Were they label-noisy data? Additionally, information on the pretraining performance, such as accuracy, is needed.

3. The proposed method is only evaluated on multi-class classification problems, where a single label is predicted for a given image. Could you explain whether the proposed method can be applied to multi-label classification problems, where multiple labels are predicted?

4. It is important to consider whether there is a constraint on the noise level that could negatively impact the proposed method's performance. Although the experiment section shows different levels of noise, it would be beneficial to indicate the threshold beyond which the proposed method fails to handle label noise effectively.

---

> ### Author Rebuttal · Authors · 2025-07-29
>
> We sincerely appreciate the time and effort the reviewer has dedicated to reviewing our work.
>
> **About Eq (3)**: Equation (3) is intentionally designed to focus on classes whose predicted probability from the k-nearest neighbor aggregation is equal to or greater than that of the original label because these are the classes that actively compete with the current label in terms of likelihood. Our goal is to estimate the likelihood that the current label is incorrect, and classes with lower predicted probabilities are irrelevant to that decision, as they do not offer sufficient evidence to challenge the existing label. Including classes with lower scores would dilute the signal and introduce noise into the correction probability. By normalizing over only the competitive classes, the policy precisely quantifies the relative disagreement strength and produces sharper, more meaningful gradients for learning.
>
> **About pretraining $f_\theta$** : The feature extractor $f_\theta$ is initially pretrained on the same label-noisy dataset using standard cross-entropy (CE) loss, without access to any clean supervision. This design choice reflects a realistic and widely adopted assumption in noisy-label learning, where no trusted labels are available at initialization. Despite the presence of label noise, this pretraining stage enables $f_\theta$ to learn useful semantic representations that serve as a strong foundation for the subsequent reinforcement learning-based label correction policy. The performance of this pretrained model matches the 'CE' baseline reported in our experimental tables. For clarity and convenience, we include those results below:
>
> Pretraining Accuracy of $f_\theta$ (CE Baseline) on different datasets and noise rates
>
> | Dataset        | Noise Rate | Accuracy (%) |
> |----------------|------------|---------------|
> | CIFAR10-IDN    | 0.20       | 75.8          |
> |                | 0.30       | 69.2          |
> |                | 0.40       | 62.5          |
> |                | 0.45       | 51.7          |
> |                | 0.50       | 39.4          |
> | CIFAR100-IDN   | 0.20       | 30.4          |
> |                | 0.30       | 24.2          |
> |                | 0.40       | 21.5          |
> |                | 0.45       | 15.2          |
> |                | 0.50       | 14.4          |
> | ANIMAL-10N     | --         | 79.4          |
> | Food-101N      | --         | 81.6          |
>
>
>
>
> **About multi-label prediction:** Our proposed method is specifically designed to address the standard learning with noisy labels setting under multi-class classification, where each instance is associated with a single label. Extending RLNLC to the multi-label setting, where multiple, potentially noisy labels may be associated with a single instance, introduces new challenges, such as modeling inter-label dependencies and defining appropriate action and reward formulations for partially correct label sets. While our current formulation does not directly support this scenario, we view multi-label noisy label correction as a promising direction for future work..
>
> **About constraints on the noise level:** There is no fixed or universal failure threshold for RLNLC, as its performance depends not just on the overall noise rate, but on the interplay between class separability and the local structure of label noise, specifically, how noise affects neighborhood consistency in feature space.  RLNLC leverages local agreement in the embedding space and optimizes label corrections via long-term reward feedback. This enables it to remain effective as long as a non-trivial fraction of each sample’s neighbors retain correct labels. Empirically, we observe that even at 90% CCN on CIFAR-100 in Table 5, RLNLC degrades gracefully and continues to outperform strong baselines.

---

> > ### Comment · Reviewer_uqZQ · 2025-08-07
> >
> > Thank you to the authors for their detailed response, which addressed  my questions. Therefore I keep my original rating.

---

### Comment · Area_Chair_wwwb · 2025-08-06

Dear Reviewer,

Thank you for your time and expertise in reviewing for NeurIPS 2025. As we enter the discussion phase, we kindly encourage you to read and respond to authors' rebuttals and clarify any outstanding issues. Your input is important to ensure a fair review process.

Please take a moment to share your thoughts when convenient. Active and timely engagement in the discussion phase is highly appreciated.

Please don’t hesitate to reach out if you have any questions.

With gratitude,

Your AC

---

### Decision · Program_Chairs · 2025-09-17

**Decision:**

Accept (poster)

**Comment:**

This paper proposes a reinforcement learning framework for noisy label correction, modeling the task as an MDP with well-defined state, action, and reward components. The method features a novel reward design based on label consistency and neighborhood alignment, and is trained via an actor-critic strategy.

The paper received mostly positive reviews. Reviewers appreciated the originality of modeling label correction as a sequential decision problem, the clarity in defining the MDP components, and the experimental validation. All reviewers acknowledged the potential impact of

Most concerns were adequately addressed during the discussion phase, and the overall contribution remains technically sound, original, and practically relevant. I recommend acceptance.